# Multisource Smart Computer-Aided System for Mining COVID-19 Infection Data

**DOI:** 10.3390/healthcare10010109

**Published:** 2022-01-06

**Authors:** Mohammad T. Abou-Kreisha, Humam K. Yaseen, Khaled A. Fathy, Ebeid A. Ebeid, Kamal A. ElDahshan

**Affiliations:** Mathematics Department, Faculty of Science, Al-Azhar University, Cairo 11651, Egypt; mt@alazhar.edu.eg (M.T.A.-K.); Khaledfathy@azhar.edu.eg (K.A.F.); ebeid78@azhar.edu.eg (E.A.E.); dahshan@gmail.com (K.A.E.)

**Keywords:** computer-aided diagnosis (CAD), COVID-19, data mining, deep learning, diagnosis, machine learning, medical information system, transfer learning

## Abstract

In this paper, we approach the problem of detecting and diagnosing COVID-19 infections using multisource scan images including CT and X-ray scans to assist the healthcare system during the COVID-19 pandemic. Here, a computer-aided diagnosis (CAD) system is proposed that utilizes analysis of the CT or X-ray to diagnose the impact of damage in the respiratory system per infected case. The CAD was utilized and optimized by hyper-parameters for shallow learning, e.g., SVM and deep learning. For the deep learning, mini-batch stochastic gradient descent was used to overcome fitting problems during transfer learning. The optimal parameter list values were found using the naïve Bayes technique. Our contributions are (i) a comparison among the detection rates of pre-trained CNN models, (ii) a suggested hybrid deep learning with shallow machine learning, (iii) an extensive analysis of the results of COVID-19 transition and informative conclusions through developing various transfer techniques, and (iv) a comparison of the accuracy of the previous models with the systems of the present study. The effectiveness of the proposed CAD is demonstrated using three datasets, either using an intense learning model as a fully end-to-end solution or using a hybrid deep learning model. Six experiments were designed to illustrate the superior performance of our suggested CAD when compared to other similar approaches. Our system achieves 99.94, 99.6, 100, 97.41, 99.23, and 98.94 accuracy for binary and three-class labels for the CT and two CXR datasets.

## 1. Introduction to COVID-19 and Diagnosis

The widespread COVID-19 pandemic constitutes a severe threat to global health. Therefore, most new research has used tools and techniques for tracking COVID-19 and discovering various infection areas to minimize the risk of its spread. Because of the massive quantity of data available every day for COVID-19 infection, spread, detection, deaths, etc., there is a need for big data analytics, storage, and security in NoSQL database management systems [1,2]. Machine learning and AI approaches can evaluate large quantities of COVID-19 data to create new models and techniques for diagnosing COVID-19. Big data analysis techniques are crucial to analyze more data in less time, as time is a critical factor for treating COVID-19 infection cases. Furthermore, AI techniques enable a global visualization of the analyzed big data of COVID-19. The visualization uses AI to present an overview of global health and confirmed cases of COVID-19. In addition, the presented images of the lungs can indicate the presence of COVID-19. Therefore, the tracking of COVID-19 diseases to enhance community health needs comprehensive data and intelligent computational instruments. In a variety of approaches, numerous researchers have employed big data and AI tools to track COVID-19 disorders, as shown in Figure 1.

COVID-19 is an infectious disease where coronaviruses are a large family of viruses that can affect both humans and animals and cause respiratory difficulties [1]. Historically, 2020 was considered a volatile year for humans worldwide compared to previous years because of COVID-19, as it is a massive threat to global health. As of March 2021, there have been more than 128 million confirmed illnesses and approximately 3 million deaths worldwide [2]. Therefore, the number of infected subjects is increasing, with more than 150 countries reportedly having at least one case [3].

Image scanning is helpful to diagnose COVID-19 for infected subjects. Patients that have been exposed and have terrible symptoms of the virus may not be identified by the outcome of RT-PCR tests [4,5,6] that can still be non-deterministic. Image scanning includes X-rays (CXR), and computed tomography (CT) images. CT scans have proven to be one of the most accurate methods of diagnosis for COVID-19 [7]. However, there are several significant drawbacks [8], such as the high cost and not being conducive to bedside testing [9]. Consequently, it is not usually used in COVID-19 diagnosis, and it is also not necessary for the progression of specific cases to be observed, especially in seriously ill patients [10]. On the contrary, the X-ray technique is a less sensitive method than CT for COVID-19 detection, with a reported baseline hypersensitivity of 69 percent [11]. The X-ray is also a cheaper, faster option and can be used in many healthcare centers. Positive X-ray results reduce the need for CT screening if there is a strong clinical suspicion of COVID-19 infection [11]. However, this presents limitations for patients, including pregnant women, since it can affect the fetus [12]. In both lungs, radiologists also examine multiple patchy, segmental, or sub-segmental shadows in the ground glass density when analyzing X-ray images to diagnose COVID-19 [13]. This can be automated to assist experts in making a decision [14,15,16].

Therefore, big data and AI technology offer an essential role in the battle against COVID-19. Both tools might help doctors to diagnose COVID-19 cases more quickly and accurately. Accordingly, computer-based models for predicting, foretelling, analyzing, and distributing SARS-CoV-2 drugs have been designed and developed, allowing machine learning, computer vision, and robotic technology to be applied. In addition, AI and big data tools include visualization to illustrate information that supports regional transmission and risk allocation.

Different studies [17,18] were carried out based on in-depth learning technologies to diagnose and classify various diseases, such as viral pneumonia and organ tumors. Today, deep learning technologies have been used widely in the healthcare domain. This study makes the following contributions:A deep learning sample-efficient algorithm for the diagnosis of COVID-19 based on CXR and CT scans.Three COVID-19 datasets were used to train and test the proposed CAD. The datasets include 4001 positive CT scans of COVID-19 clinical results and 3835 positive CXR images. It is the most widely accessible CT dataset for COVID-19 as far as we are aware.An extensive analysis of the results of COVID-19 transition was planned and conducted; informative conclusions are presented through developing various transfer techniques.Self-controlled learning with transfer learning was utilized to learn strong, impartial representations of features to reduce the chance of over-fitting to learn from restricted labeled information.Detailed studies were carried out to show that our CAD was successful. The results, on average, were a 99.18% accuracy, 99.69% recall, and 99.4% precision on the COVID-19 CXR and CT imaging datasets.

This paper discusses the most recent research in Section 2. Section 3 presents an overview of the methods and techniques used. The proposed model is presented in Section 4. Section 5 gives a brief description of the dataset used and explains the computer system configuration, parameter settings, and performance metrics. Section 6 presents the experiment and discussion. Finally, Section 7 concludes the paper with an outline of future work.

## 2. Background on Machine Learning and Deep Learning

Deep learning (DL) is a subset of the machine learning (ML) branch, the third generation of artificial neural networks. The principal objective of DL is the simulation of high-level data abstractions [19,20,21]. Different DL utilizes numerous layers to remove upper-level functions progressively from the raw data. DL produces several neuron layers, organized layer per layer.

For computer vision and image processing, there are numerous architectures of various types, such as generative adversarial networks (GANs) [22], convolutional neural networks (CNNs) [23,24], and DE convolutional networks [25].

CNNs are mainly utilized for images. CNNs are new deep learning algorithms suggested by Badrinarayana [24]. CNN lines distinguish among the weights of different artifacts in the image. This approach needs less pre-processing comparing with other shallow classification algorithms [26]. For input images, a CNN uses filters to capture spatial and time dependencies [27]. In CNNs, the height, m, and width, n, and r correspond to the channel number or depth. The input is separated by m and r instead of the three input components, m×m×r. There are several kernels of size *k* in every convolution layer [28]. As mentioned previously, the filtering is the base of relations, along with the development of k maps of each size (*m*, *m*, 1), each with the same parameters. The convergence layer calculates the point product, similar to MLP, among weights and inputs, except for a small amount of the original volume of the input, as shown in Equation (1). In addition, an activation function for the non-linearity function activates the contribution of the convolutional layers [27]:(1)$$h^{k}=f\left(W^{k}*s+b^{k}\right)$$

The output of the current k-layer is denoted by $h^{k}$, the kernel or weight of the current layer is indicated by $W^{k}$, s presents the output of the previous layer, and $b^{k}$ represents the current bias of the current layer. The number of computational parameters is an essential indicator of a deep learning model’s complexity. The output characteristic maps can be described according to the following formula [27]:(2)$$M=\frac{\left(N-F\right)}{S}+1$$

The input map dimensions are denoted by *N* and filter dimensions or receptive area by *F*, while *M* refers to output map dimensions and *S* to the stride length. Usually, padding is used to guarantee input and output during convolution operations that are the same size.

The padding number varies according to the kernel size. The number of rows and columns for padding is calculated in Equation (3) [29].
(3)$$P=\frac{\left(F-1\right)}{2}$$
where the amount of padding is symbolized by P, and *F* represents the dimensions of the kernels.

One of the most important principles in computer engineering is the reusability of components. In turn, many architectures have been introduced, including AlexNet, ResNet-50, ResNet-101, VGG-16, and VGG-19 [27,30]. Therefore, we intend to reuse the model regards to transfer learning guidelines. The transfer learning process reuses information from the source domain in the target domain [31]. See Figure 2 for extra explanation. Parameter optimization, structural reformulation, regularization, etc., are different improvement categories that were interested by many research communities. However, the main drive in CNN performance improvement appears to have come from the rearrangement of processing units and the design of new blocks. The majority of advancements in CNN designs have been carried out in the areas of depth and spatial exploitation to develop an excellent internal representation from raw pixels without requiring extensive processing.

AlexNet is considered as a type of feed-forward CNN with depth of eight layers and a spatial exploitation architecture [32]. It has five convolution layers (conv1 through conv5) as well as three completely connected layers (fc6, fc7, fc8) [33]. It was trained by classifying 1 million photos into 1000 different categories [23].

VGG-16 was trained using the same training set used for AlexNet. It contains three fully connected layers (fc6, fc7, fc8) and five convolutional blocks comprising 13 convolutional layers [34]. On the other hand, VGG-19 comprises 19 layers, including five convolutional blocks of 16 convolutional layers and three fully connected layers (fc6, fc7, and fc8).

Each ResNet type, such as ResNet-50 and ResNet-101, has its residual block. ResNet-50 is a 50-layer network that is cascaded from a convolution layer to 16 residual blocks within the network and finally to a fully linked layer. ResNet-101 has a total of 101 layers and 33 residual blocks [35]. Table 1 shows how contemporary models compare in terms of error, network parameters, the maximum number of connections, and more.

Machine learning (ML) algorithms are known for learning underlying relationships in data and making decisions without the need for explicit instructions. The capacity of a CNN to utilize spatial or temporal correlation in data is one of its most appealing features. A support vector machine (SVM) is a shallow classification algorithm developed by Vapnik [36]. The SVM is classification algorithm reduces learning steps and offers a quicker solution than other common algorithms [37,38]. The SVM classifier is built on the concept of the most appropriate hyper-planes, which are used to differentiate between two classes, positive or negative as shown in Equation (4) [39] by including the central function.
(4)$$f\left(x\right)=sign((\sum_{i=1}^{n}\alpha_{i}y_{i}K\left(x_{i},x\right))+b)$$

## 3. Brief Coverage of Previous Works

Many researchers are currently encouraged to establish early detection models to detect COVID-19 infection before outbreak:

Zhou Tao et al. [40] proposed EDL_COVID (an ensemble deep learning model) to detect COVID-19 disease from 2933 CT images. The proposed model depends on the three ensemble models AlexNet, GoogleNet, and ResNet.

An ensemble strategy was proposed by Rohit Kundu et al. [41] for detecting COVID-19 in CT scan images for human lungs. They employed two datasets of CT scan images to create decision scores for the proposed ensemble model utilizing three CNN models: VGG-11, ResNet-50-2, and Inception v3.

The authors proposed a deep convolutional 3D neural network called DeCoVNet to identify COVID-19 from CT images [15]. Thus, when COVID-19 was diagnosed, the algorithm worked in a black box because it focused on DL and was still at an early stage of explanatory ability.

COVNET [16] has developed and tested the efficiency of COVID-19 detection utilizing chest CT. The researchers have proposed a 3D deep learning system. The robustness evaluation of the model included community-acquired pneumonia (CAP) and other non-pneumonia exams.

In contrast with the RT-PCR assay of COVID-19, Yang et al. [18] assessed the agnostic and consistency value of chest CT. They suggested that chest CT should be considered, particularly in epidemic areas with a high preliminary possibility of disease for screening of COVID-19, comprehensive assessment, and follow-up.

Horri et al. [32] used three different methods of physician imaging (X-ray, ultrasound, and CT) for diagnosing COVID-19 stably and automatically. They utilized a deep VGG transmission learning network to refine their analysis. The accuracy of their classification was stated to be 86 percent, 84 percent, and 100 percent for three different datasets.

Ying et al. obtained a 94% accuracy and a 99% AUC with CT images utilizing a deep model based on ResNet50, known as DRE-Net [42]. They also considered an approach for target identification, i.e., indicating the areas of concern with bounding boxes [43]. VGG architecture [44] has been used to diagnose symptomatic lung regions [34]. A suggested method distinguishes cases of pneumonia (CAP) and non-pneumonia from COVID-19 (NP) in the population.

Jiang et al. [15] proposed an early screening strategy using pulmonary CT imaging to distinguish COVID-19 mutations from viral influenza pneumonia and stable cases. Several CNN models were suggested and utilized to identify the CT image datasets and quantify the risk of infection with COVID-19. The results may be beneficial in deep learning technologies for the early screening of COVID-19 patients. In the classic ResNet for feature extraction, the authors have proposed a location-attention mechanism.

The AIMDP model was suggested [42] for use with mutable artificial intelligent techniques to improve the model’s diagnosis and predictive role. The authors [32] developed a framework focused on the deep learning of the detection of CT viral pneumonia.

The authors in [44] also provide an overview of the most recent artificial intelligence systems in X-ray images for COVID-19 diagnostics. However, they used X-ray images, as their work was based on this only. To estimate COVID-19 diagnostics, Ghoshal et al. [45] presented a Bayesian convolutional neural network, differentiating between COVID-19 and non-COVID-19 cases, with a 92.9 percent accuracy. Binary classification was carried out by Narin et al. [46] for detecting COVID-19 to achieve the best accuracy of 98.0% with ResNet50 models, compared to the various deep learning (DL) models. Zhang et al. [47] submitted the COVID-19 (0.952 AUC) ResNet model to illustrate the pneumonia areas affected by applying the Grad-CAM approach for the gradient activation.

Finally, Wang et al. [48] suggest a deep CNN rated as 83.5 percent accurate between the VCOVID-19, non-COVID-19, and uninfected cases.

These studies have provided detailed solutions for combating the pandemic COVID-19. However, there are certain drawbacks to be taken into account. In the best case, researchers used small datasets of fewer than 400 images of COVID-19. In some cases, only 10 X-ray images were used for the COVID-19 class to validate the framework. Furthermore, there was no ground for comparison or medical surveillance with the obtained results, which can suggest not only COVID-19 identification but also the location of influenced areas in the lungs. For iteratively sliced COVID-19 identification using X-ray pictures, a deep learning model ensemble is proposed [49]. This research made use of a CNN and a set of pre-trained models. The proposed algorithm enhances memory efficiency while reducing complexity.

## 4. Architecture of the Smart CAD System

The proposed CAD system depends on deep learning, transfer learning, and shallow machine learning. In deep learning, multi-hidden layers are stacked for learning objects. These layers require a training process including “fine-tuning” to slightly adjust the weights of the DNN found in pre-training during the backpropagation procedure. Hence, DL nets can extract, classify the features, and effectively make a precise decision after an efficient training process. Transfer learning is used in the proposed CAD system to optimize multiple CNN architectures for datasets. However, the transfer-learning methodology generates optimal fitted CNNs for the datasets capable of classifying and diagnosing infection of COVID-19 scan images. In addition, these fine-tuned models can extract the feature set usable by the different shallow classifiers. Figure 3 shows a context diagram starting from scanning the image of the inspected case until detecting the infection response using the proposed smart CAD system, in which there are key components comprising the proposed CAD system, including:
Scanning: The source of the input image used to check the status of COVID-19 infection. The supported format of scans can be either CT or CXR images.Pre-processing: A set of procedures performed for every newly scanned image before investigating the diagnosis process. It comprises auto color correction, auto contrast enhancement, resizing the image to the standard size, and normalizing color channels.Diagnosing: A key component of the medical CAD system to detect, assist, and advise the doctors in their inspection and symptom analysis during the examination process. It can be divided into:
✓Classification: A vital component of the smart CAD system in which different architectures can be alternately used. These models are responsible for extracting the features and the classification.✓Decision Unit: This depends on the most common and powerful DL activation function, ReLu. It is a subsequent responsibility of the classification component to make a decision.

Figure 4 shows different phases of the proposed system in a layered sub-black box style, in which the essential layers are briefly described for the proposed smart CAD system. According to current knowledge, all COVID-19 detection systems consist of a few significant layers: input data, model layer, activation layer, and model layer output for CXR or CT scan image analysis. In turn, the classification and decision in every CAD system using deep learning must include a collection of these different layers. Each group of these layers with a specific order is called a network architecture starting from input layer to output layer (e.g., AlexNet, VGG-16, VGG-19). Next, a brief description defining each role is discussed in detail along with its importance for the medical CAD system.

### 4.1. Input Layer

This layer reads the image data collection in advance. In other words, the CXR and the CT scan images are pre-processed independently. In the pre-processing phase, the images are reconstructed and resized. The images are taken from various sources, and their dimensions vary since the taken images from medical instruments were created from several letters, arts and crafts, and medical symbols. Moreover, the model layer of each of these products needs separate image dimensions to be managed. Therefore, the input image size was adjusted to fit the templates used in this analysis rather than cutting the lung and chest area as far as possible.

### 4.2. Model Layer

This layer represents the leading layer of the proposed smart CAD system, in which most calculations are carried out. The calculations include extracting image dataset features and preserving the spatial relationship between image pixels. Next, the data are moved from the input layer to the model layer. This layer contains four sub-black boxes. The CNN-based AlexNet was used with the aim of utilizing AlexNet’s pre-trained approach to diagnose COVID-19. The second sub-layer is the CNN-based RESNET of two versions RESNET50 and RESNET101, distinguished from other architectures by adding to the model blocks that feed the values into their following layers. This value changes the device value as described by adding a block every two layers between the linear and the ReLu activation codes. However, ResNet101 architecture uses more layers than the blocks of ResNet50 with three layers. The ResNet50 model offers fast training and considerable benefit because image residuals are learned rather than functionality [35].

The third sub-layer is the VGG sub-layer based on the CNN. Although it is a single model, the main advantage related to previous versions is that the CNN models are commonly used, so they are organized more thoroughly and accompanied by two- or three-color layers. VGG has a strong representation of features, and the model can serve as a helpful extractor for new images [34]. The last sub-layer is the SVM classification. Since the SVM is a good classification algorithm, it can be used to classify features that have already extracted. The methods used for feature extraction were derived from previous sub-layers (see Figure 5).

### 4.3. Activation Layer

This layer is a non-linear map of CNN architectures that works at the end of the learning phase to replace negative pixel values with zero in the convolved functions.

### 4.4. Output Layer

Based on the output score of the activation layer, the final response of classification is provided as an output label. The resulting label can be numerically categorized or encoded; for example, “0” is marked with COVID-19 (i.e., the positive event), “1” is marked with regular cases, and “2” is marked with other cases of pneumonia, etc.

## 5. Experimental Result

The proposed model was evaluated in-depth to assess the efficiency of the solutions and examine the impact on transfer learning and self-controlled learning. In the following subparagraphs, we describe the utilized datasets for the proposed CAD system, experimental environment, settings, and results due to performance metrics.

### 5.1. Dataset Description

Three datasets were used in these experiments, two of which have images of CXR type, and the last has CT images. The acquired dataset of CT scans was divided into 4001 COVID-19 and 15,684 non-COVID-19 images, whereas the first CXR dataset consists of 219 COVID-19 and 2686 non-COVID-19 images. The second CXR dataset comprises 3616 COVID-19 and 17,549 non-COVID-19 images. The evaluation supports the holdout procedure using 80% training set and 20% testing set. See Table 2 for a brief description of dataset details. Figure 6 and Figure 7 show a montage preview of the CT and CXR images.

### 5.2. Experimental Details

#### 5.2.1. Computer System Configuration

The proposed CAD system was implemented using MATLAB R2020a, computer vision, image processing, neural networks, and deep learning toolboxes. The CAD system works on a HP Zbook workstation with Windows 10 64-bit, CPU: i7-6820HQ, RAM: 32GB DDR5, and GPU: 8 GB.

#### 5.2.2. Parameter Settings

All networks were trained as follows: optimizer, SGDM initial learning rate 0.0001, and validation frequency 5. Every epoch, which is a complete cycle of training iteration, in the dataset was shuffled, and the training process stopped if the it did not change significantly. For all networks, the dataset was divided into 80% and 20% for training and validating sets, respectively. For all networks, the same training and validation datasets were chosen to facilitate the performance comparison of networks.

#### 5.2.3. Performance Metrics

For the proposed CAD system, there are different performance metrics for evaluating efficiency and effectiveness. In such cases, the negative and positive cases were assigned to the non-COVID-19 and COVID-19 infection groups, respectively. In sequence, the number of correctly detected COVID-19 and non-COVID-19 infections is represented by $N_{\mathrm{TP}}\;\mathrm{and}\;N_{\mathrm{TN}}$, respectively, whereas $N_{\mathrm{FP}}\;\mathrm{and}\;N_{\mathrm{FN}}$ indicate the number of incorrectly diagnosed COVID-19 and non-COVID-19 infections, respectively. Table 3 represents a brief description of the most common metrics used for evaluating the proposed CAD system.

## 6. Experiment Design: Result Evaluation and Discussion

The proposed CAD system was evaluated using two scenarios per single dataset; hence, six experiments were performed. The first scenario depends on optimizing parameters and fine-tuning pre-trained networks as an end-to-end CAD component. The second scenario involves employing the developed component in the first scenario as a feature extractor engine. The feature extractor engine then passes these feature sets to an SVM classifier boosted by optimizing the kernel function as a hybrid learning CAD component. The recorded results were captured per the dataset regarding the stated two scenarios, and the most effective model of the dataset was determined. In the following, the results are divided into three subsections for each dataset.

### 6.1. CT Scan Dataset

Firstly, the experiments were started with CT scan images, and as mentioned above, there are two scenarios for each dataset. The first scenario is exhibited here with the two-class label as the normal state and COVID-19. The same scenario was performed for the three-class dataset.

Table 4 and Table 5 show the numerical results for the first scenario for the CT scan dataset. Table 4 and Table 5 show all the results for the proposed models with three metrics, namely accuracy, precision, and recall, for each of the fully deep learning and the hybrid learning solutions. The experimental analysis shows the superiority of the proposed models over various metrices. Therefore, the same experiment was repeated for two new datasets for the X-ray images, as outlined in the following two subsections.

### 6.2. The First X-ray Dataset

As shown in the previous subsection, our proposed models are either fully deep learning or a hybrid model with the SVM by applying the first X-ray dataset. This experiment starts with the two-class label then the three-class label as discussed before. Table 6 and Table 7 show the results for the two-class label and three-class label, respectively.

In this experiment, the two classes’ dataset hybrid VGG19-SVM shows the best performance measures compared to both the other models. Even with the three-class dataset, the fully deep learning (enhanced TL of VGG16) method gives better results for accuracy and recall than the hybrid learning solutions.

### 6.3. The Second X-ray Dataset

Lastly, the two scenarios were applied for the second X-ray dataset. This experiment’s results are shown in Table 8 and Table 9 for the two-class label and the three-class label, respectively.

## 7. Discussion

This section discusses the superiority of the proposed models versus the related models in recent literature studies. The proposed model has multi-source scan images based on modularity, including CT scan and X-ray images. First, for the CT scan, Table 10 shows our proposed model deducted from the comparative study in Table 4 and the literature for the same dataset and inputs—results of this experiment are visualized in Figure 8. The second scenario was performed on the three-class label for the same dataset. All comparative results were replicated according to the three-class label dataset, as shown in Table 11 and Figure 9. In turn, the experiments were carried out in which the end-to-end VGG16 with the binary class demonstrated its superiority to the hybrid model. With three classes, the hybrid model achieved better results, and both showed better results than the comparative study from the literature.

Second is the first X-ray dataset where the proposed model obtained accuracy lower than Muhammed E.H. et al. [51] of around 0.89%, and the proposed model achieves a much more reasonable recall rate. Consequently, our proposed model does not stick in under-fit or over-fit with regards to a specific label (see Table 12). The proposed model satisfies the balance classification rates between different labels in the given dataset. Furthermore, the proposed model achieves a notable enhancement compared to others in terms of accuracy, precision, and recall by a significant rate as illustrated in Figure 10 for binary classification. Table 13 and Figure 11 demonstrate the superiority of the proposed model versus the models in the literature for three classes.

In the third dataset, the hybrid learning solution provided better results than fully deep learning. For the binary label, the proposed enhanced TL VGG16+SVM demonstrated its superiority (see Table 14). Figure 12 represents the visual analysis of the proposed model for binary classifier in terms of accuracy, precision, and recall. The proposed enhanced TL VGG19+SVM showed its effectiveness for the three-class label dataset (see Table 15). Figure 13 shows a graphical bar chart analysis of the proposed model versus the models in the literature; both binary and multiclass models show improvements in accuracy compared to those in the literature.

## 8. Conclusions

This paper proposes a CAD system for detecting COVID-19 infection. An excellent diagnostic performance was demonstrated in using both CT and CXR images. In addition, the CAD system is superior to those found in the literature. The CAD system could be a supplementary reliable analysis tool for diagnosing COVID-19 cases using CXR and CT images. Visible features in CT scan images, such as the intensity, shape, size, and nodule margins, may influence the diagnostic efficiency of the CAD system. Furthermore, junior radiotherapists lacking experience can use these helpful suggestions provided by the proposed CAD system.

## Figures and Tables

**Figure 1 healthcare-10-00109-f001:**
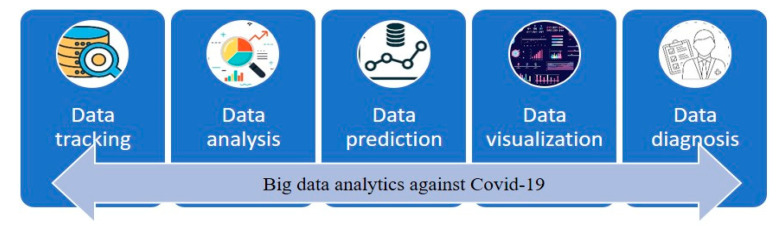
Big data analytics against COVID-19.

**Figure 2 healthcare-10-00109-f002:**
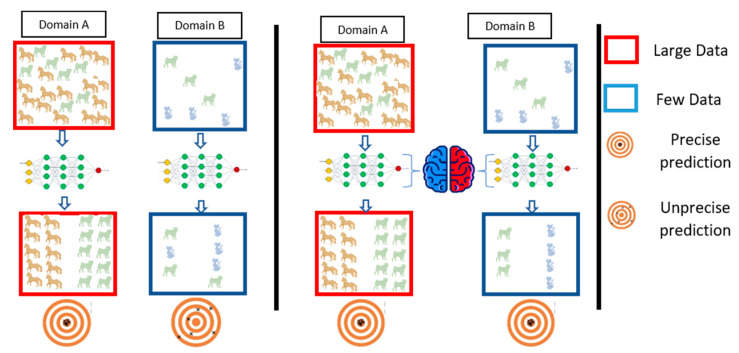
Simple overviews showing model forecasts of domain A and domain B transition (**right**) or without (**left**). The transfer learning extracts the features from domain A as common knowledge and then uses common knowledge to forecast model B.

**Figure 3 healthcare-10-00109-f003:**
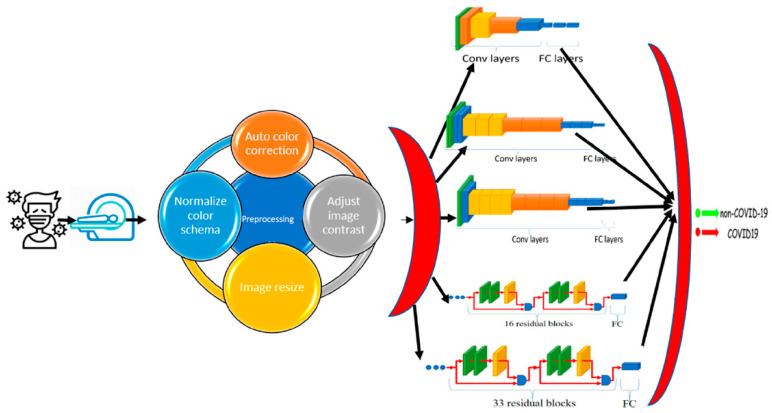
Context overview of the proposed smart CAD system.

**Figure 4 healthcare-10-00109-f004:**
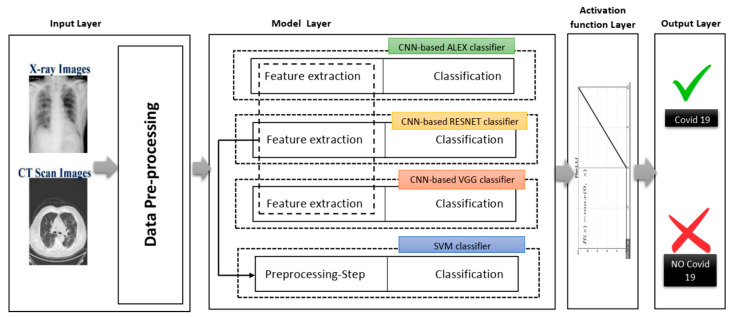
Layered phase of smart CAD system.

**Figure 5 healthcare-10-00109-f005:**
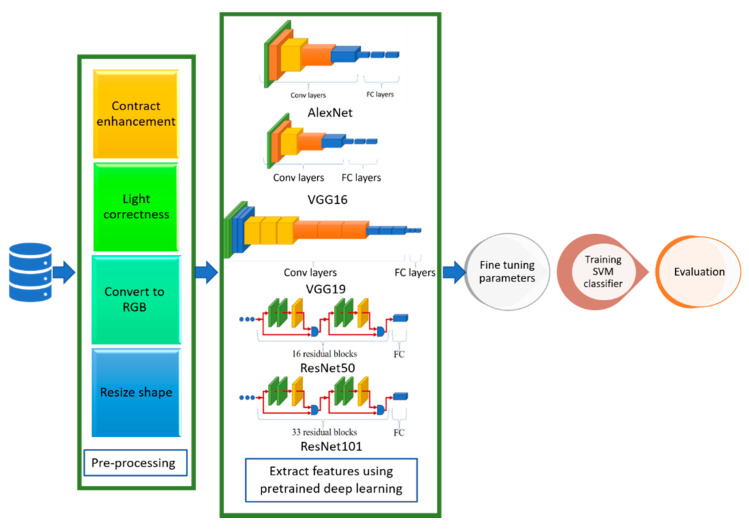
Flowchart of the hybrid learning models.

**Figure 6 healthcare-10-00109-f006:**
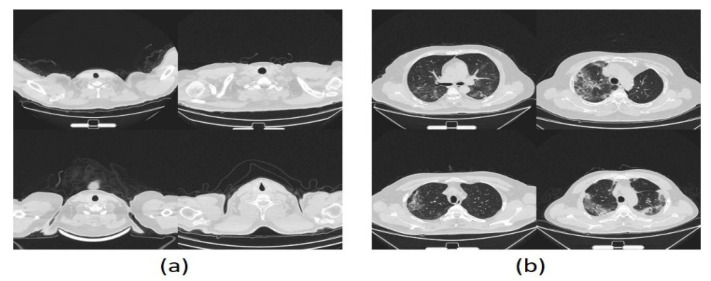
Sample of CT dataset collection: (**a**) normal cases (**b**) COVID-19 cases.

**Figure 7 healthcare-10-00109-f007:**
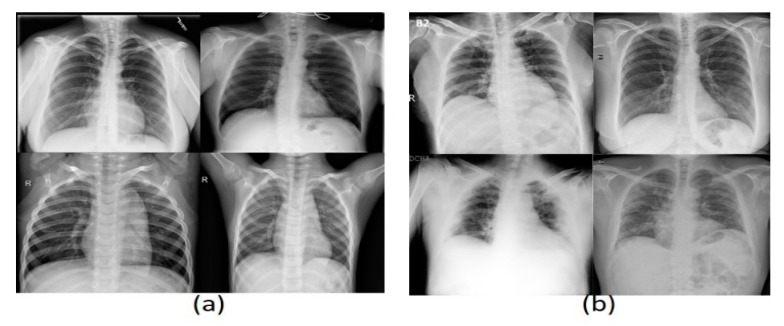
Sample of CXR dataset collection: (**a**) normal cases (**b**) COVID-19 cases.

**Figure 8 healthcare-10-00109-f008:**
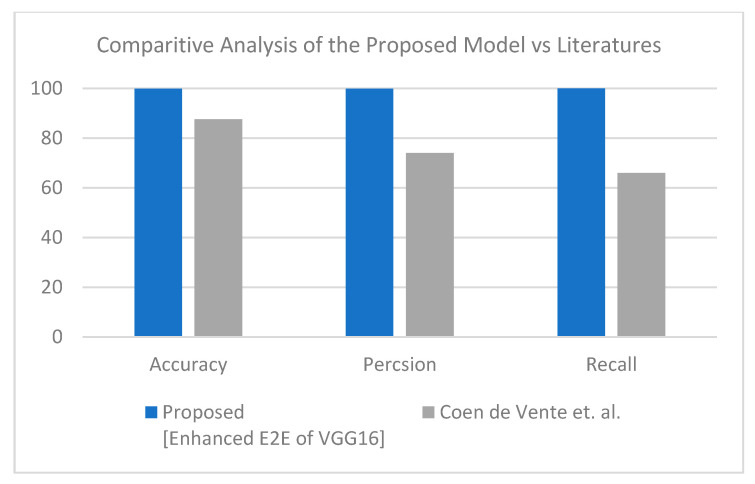
Comparative study between proposed model and other literature models using CT dataset (two-class) [50].

**Figure 9 healthcare-10-00109-f009:**
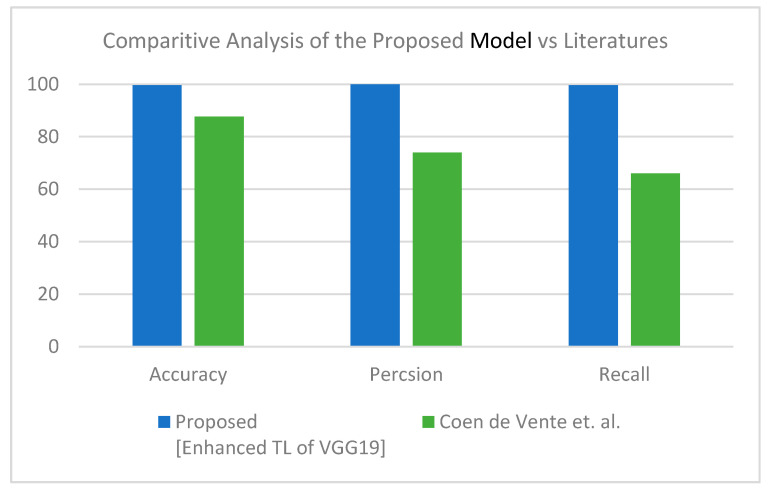
Comparative study between proposed model and other literature models using CT dataset (three-class) [50].

**Figure 10 healthcare-10-00109-f010:**
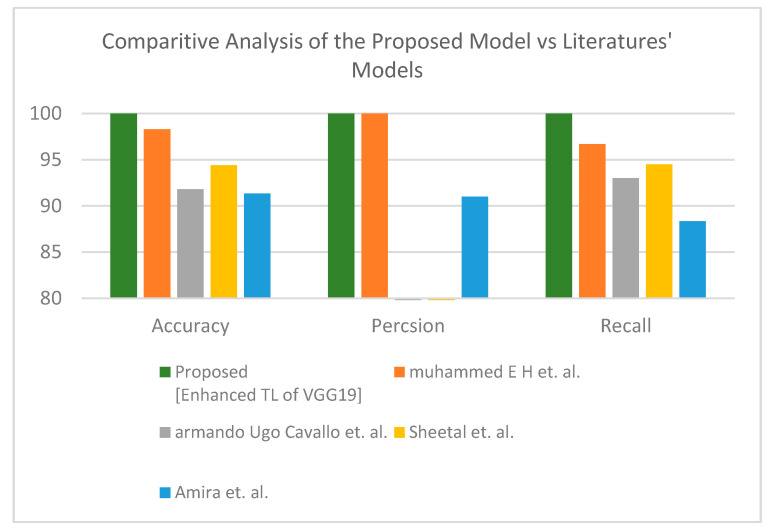
Comparative study between proposed model and other literature models using X-ray dataset (two-class) [51,52,53,54].

**Figure 11 healthcare-10-00109-f011:**
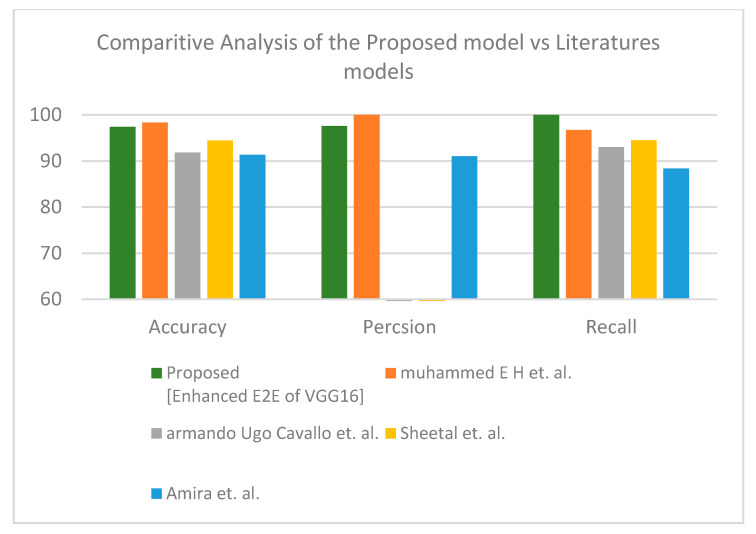
Comparative study between proposed model and other literature models using X-ray dataset (three-class) [51,52,53,54].

**Figure 12 healthcare-10-00109-f012:**
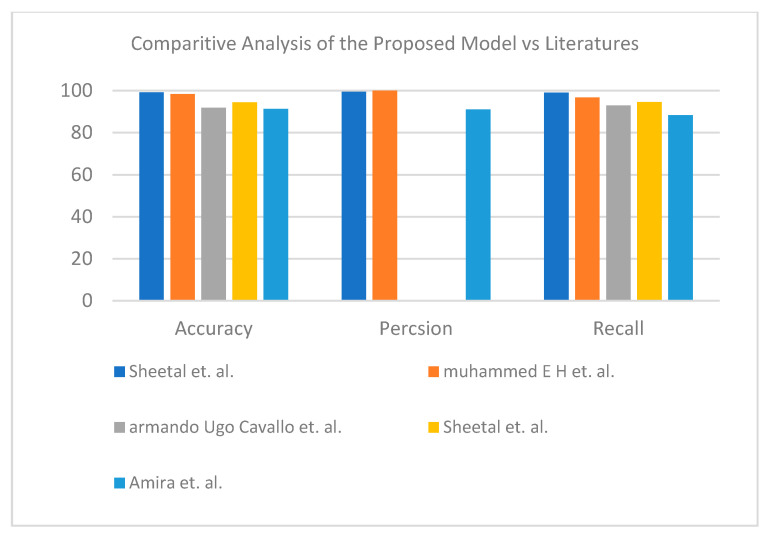
Comparative study between proposed model and other literature models using X-ray dataset (two-class) [51,52,53,54].

**Figure 13 healthcare-10-00109-f013:**
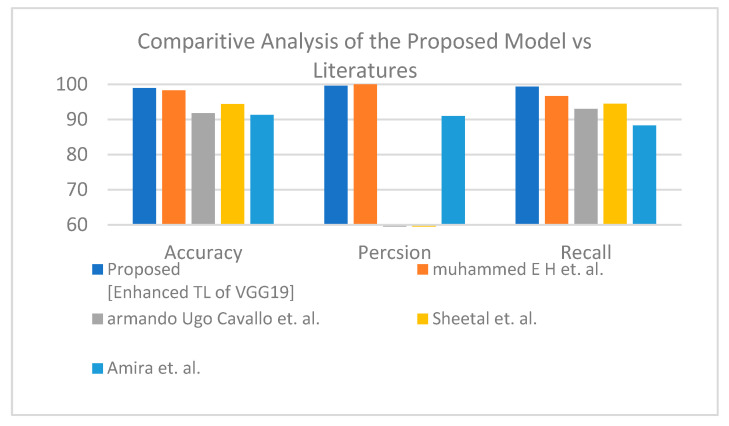
Comparative study between proposed model and other literature models using X-ray dataset (three-class) [51,52,53,54].

**Table 1 healthcare-10-00109-t001:** Characteristics of convolutional neural networks used in this study.

Model	VGG	ALEXNET	RESNET
SIZE OF INPUT	224 × 224	227 × 227	224 × 224
STRIDE	1	1.4	1.2
NO. OF FC LAYERS	3	3	1
TOP FIVE ERRORS	7.4	16.4	5.3
NUMBER OF MACS	15.3 M	666 M	3.86 G
NUMBER OF FEATURE MAPS	3–512	3–256	3–1024
NO. OF CONV. LAYERS	16	5	50
NUMBER OF WEIGHTS	14.7 M	2.3 M	23.5 M
SIZE OF FILTER	3	3, 5, 11	1, 3, 7

**Table 2 healthcare-10-00109-t002:** Technical characteristics data of patients with COVID-19 and non-COVID-19 group.

Data Type	Total No. of Images	No. of Classes	COVID-19 Image No.	Other Pneumonia Image No.	Normal Image No.	Non-Informative Image No.	Lung Opacity Image No.
CT	19,685	3	4001	-	5705	9979	-
CXR	2905		219	1345	1341	-	-
	21,165	4	3616	1345	10,192	-	6012

**Table 3 healthcare-10-00109-t003:** Common performance metrics for CAD evaluation.

Metric	Formula	Description
Accuracy	$\mathrm{accuracy}=\frac{N_{\mathrm{TP}}\;+\;N_{\mathrm{TN}}}{N_{\mathrm{TP}}\;+\;N_{\mathrm{TN}}\;+\;N_{\mathrm{FP}}\;+\;N_{\mathrm{FN}}}$	Ratio of number of all correct detected cases to the total number of cases.
Precision	$\mathrm{precision}=\frac{N_{\mathrm{TP}}}{N_{\mathrm{TP}}\;+\;N_{\mathrm{FP}}}$	Number of correct detected COVID-19 cases divided by the total input number of COVID-19 infection.
Recall	$\mathrm{recall}=\frac{N_{\mathrm{TP}}}{N_{\mathrm{TP}}\;+\;N_{\mathrm{FN}}}$	Proportion of COVID-19 cases that are correctly classified as COVID-19, with respect to COVID-19 cases.
Specificity	$\mathrm{specificity}=\frac{N_{\mathrm{TN}}}{N_{\mathrm{TN}}\;+\;N_{\mathrm{FP}}}$	Proportion of negative data points that are correctly classified as normal, with respect to all normal cases

**Table 4 healthcare-10-00109-t004:** The performance measures in applying different learning models for the CT scan images dataset (two-class) where the bolded number indicates the best result among the other classification models.

Algorithm		Accuracy	Precision	Recall
Feature Extraction	Classification			
Enhanced TL AlexNet	AlexNet	99.48	99.93	99.44
Enhanced TL ResNet-50	ResNet-50	99.63	**100**	99.56
Enhanced TL ResNet-101	ResNet-101	99.79	99.81	99.94
Enhanced TL VGG-19	VGG-19	99.84	99.81	**100**
Enhanced TL VGG-16	VGG-16	**99.94**	99.94	**100**
Enhanced TL AlexNet	SVM	99.63	99.75	99.81
Enhanced TL ResNet-50	SVM	99.83	99.84	99.95
Enhanced TL ResNet-101	SVM	99.87	99.9	99.95
Enhanced TL VGG-19	SVM	99.79	**100**	99.75
Enhanced TL VGG-16	SVM	99.79	99.87	99.87

**Table 5 healthcare-10-00109-t005:** The performance measures in applying different learning models for the CT images dataset (three-class) where the bolded number indicates the best result among the other classification models.

Algorithm		Accuracy	Precision	Recall
Feature Extraction	Classification			
Enhanced TL AlexNet	AlexNet	99.365	99.55	99.78
Enhanced TL ResNet-50	ResNet-50	99.54	99.65	99.83
Enhanced TL ResNet-101	ResNet-101	99.41	99.45	**99.9**
Enhanced TL VGG-19	VGG-19	99.49	99.57	99.88
Enhanced TL VGG-16	VGG-16	99.2634	**99.97**	99.18
Enhanced TL AlexNet	SVM	98.9	99.51	99.33
Enhanced TL ResNet-50	SVM	99.51	99.6	99.83
Enhanced TL ResNet-101	SVM	99.55	99.7	99.78
Enhanced TL VGG-19	SVM	**99.6**	99.91	99.66
Enhanced TL VGG-16	SVM	98.5	99.11	99.26

**Table 6 healthcare-10-00109-t006:** The performance measures in applying different learning models for the X-ray dataset (two-class) where the bolded number indicates the best result among the other classification models.

Learning Mode	Algorithm		Accuracy	Precision	Recall
	Feature Extraction	Classification			
Fully deep learning (E2E Solution)	Enhanced TL AlexNet	AlexNet	96.38	96.95	98.62
	Enhanced TL ResNet-50	ResNet-50	96.72	97.25	99.87
	Enhanced TL ResNet-101	ResNet-101	95.35	95.8	97.1
	Enhanced TL VGG-19	VGG-19	90.7	95.96	99.24
	Enhanced TL VGG-16	VGG-16	97.41	97.6	**100**
Hybrid learning solution	Enhanced TL AlexNet	SVM	99.67	**100**	98.86
	Enhanced TL ResNet-50	SVM	99.35	98.86	98.86
	Enhanced TL ResNet-101	SVM	99.67	100	98.86
	Enhanced TL VGG-19	SVM	**100**	**100**	**100**
	Enhanced TL VGG-16	SVM	99.67	**100**	98.86

**Table 7 healthcare-10-00109-t007:** The performance measures in applying different learning models for the X-ray dataset (three-class) where the bolded number indicates the best result among the other classification models.

Learning Mode	Algorithm		Accuracy	Precision	Recall
	Feature Extraction	Classification			
Fully deep learning (E2E Solution)	Enhanced TL AlexNet	AlexNet	96.38	96.95	98.62
	Enhanced TL ResNet-50	ResNet-50	96.72	97.25	99.87
	Enhanced TL ResNet-101	ResNet-101	95.35	95.8	97.1
	Enhanced TL VGG-19	VGG-19	90.7	95.96	99.24
	Enhanced TL VGG-16	VGG-16	**97.41**	**97.6**	**100**
Hybrid learning solution	Enhanced TL AlexNet	SVM	91.4	93.92	93.22
	Enhanced TL ResNet-50	SVM	83.3	83.89	80.02
	Enhanced TL ResNet-101	SVM	81.17	81.71	82.69
	Enhanced TL VGG-19	SVM	97.2	98.37	99.01
	Enhanced TL VGG-16	SVM	92.9	97.05	93.79

**Table 8 healthcare-10-00109-t008:** The performance measures in applying different learning models for the X-ray dataset (two-class) where the bolded number indicates the best result among the other classification models.

Learning Mode	Algorithm		Accuracy	Precision	Recall
	Feature Extraction	Classification			
Fully deep learning (E2E Solution)	Enhanced TL AlexNet	AlexNet	96.59	97.62	95.78
	Enhanced TL ResNet-50	ResNet-50	98.62	**99.71**	97.65
	Enhanced TL ResNet-101	ResNet-101	98.94	99.37	98.61
	Enhanced TL VGG-19	VGG-19	98.44	99.35	97.65
	Enhanced TL VGG-16	VGG-16	98.84	98.57	**99.24**
Hybrid learning solution	Enhanced TL AlexNet	SVM	97.7	97.91	97.72
	Enhanced TL ResNet-50	SVM	98.98	98.94	98.99
	Enhanced TL ResNet-101	SVM	98.75	98.81	98.64
	Enhanced TL VGG-19	SVM	98.98	99.3	98.75
	Enhanced TL VGG-16	SVM	**99.23**	99.44	99.1

**Table 9 healthcare-10-00109-t009:** The performance measures in applying different learning models for the X-ray dataset (three-class) where the bolded number indicates the best result among the other classification models.

Learning Mode	Algorithm		Accuracy	Precision	Recall
	Feature Extraction	Classification			
Fully deep learning (E2E Solution)	Enhanced TL AlexNet	AlexNet	96.3	98.16	97.63
	Enhanced TL ResNet-50	ResNet-50	98.48	99.29	99
	Enhanced TL ResNet-101	ResNet-101	97.78	98.89	98.65
	Enhanced TL VGG-19	VGG-19	98.54	99.44	99.12
	Enhanced TL VGG-16	VGG-16	95.37	99.36	95.25
Hybrid learning solution	Enhanced TL AlexNet	SVM	97.32	98.35	98.22
	Enhanced TL ResNet-50	SVM	98.46	99.29	99.08
	Enhanced TL ResNet-101	SVM	97.91	98.83	98.64
	Enhanced TL VGG-19	SVM	**98.94**	**99.59**	**99.38**
	Enhanced TL VGG-16	SVM	98.74	99.52	99.22

**Table 10 healthcare-10-00109-t010:** Comparison between proposed model versus other related models for the CT scan dataset (two-class).

Algorithm	Accuracy	Precision	Recall
Proposed (Enhanced E2E of VGG16)	99.94	99.94	100
Coen de Vente et al. [50]	87.63	74.00	66.00

**Table 11 healthcare-10-00109-t011:** Comparison between proposed model versus other related models for the CT scan dataset (three-class).

Algorithm	Accuracy	Precision	Recall
Proposed (Enhanced Hybrid of ResNet-101 and SVM)	99.6	99.91	99.66
Coen de Vente et al. [50]	87.63	74.00	66.00

**Table 12 healthcare-10-00109-t012:** Comparison between proposed model versus other related models for the X-ray dataset (two-class).

Algorithm	Accuracy	Precision	Recall
Proposed (Enhanced TL of VGG19)	100	100	100
Muhammed E.H. et al. [51]	98.30	100.00	96.70
Armando Ugo Cavallo et al. [52]	91.80	--	93.00
Sheetal et al. [53]	94.40	--	94.50
Amira et al. [54]	91.34	91.00	88.33

**Table 13 healthcare-10-00109-t013:** Comparison between proposed model versus other related models for the X-ray dataset (three-class).

Algorithm	Accuracy	Precision	Recall
Proposed (Enhanced E2E of VGG16)	97.41	97.6	100
Muhammed E.H. et al. [51]	98.30	100.00	96.70
Armando Ugo Cavallo et al. [52]	91.80	--	93.00
Sheetal et al. [53]	94.40	--	94.50
Amira et al. [54]	91.34	91.00	88.33

**Table 14 healthcare-10-00109-t014:** Comparison between proposed model versus other related models for the X-ray dataset (two-class).

Algorithm	Accuracy	Precision	Recall
Proposed (Enhanced TL VGG16)	99.23	99.44	99.1
Muhammed E.H. et al. [51]	98.30	100.00	96.70
Armando Ugo Cavallo et al. [52]	91.80	--	93.00
Sheetal et al. [53]	94.40	--	94.50
Amira et al. [54]	91.34	91.00	88.33

**Table 15 healthcare-10-00109-t015:** Comparison between proposed model versus other related models for the X-ray dataset (three-class).

Algorithm	Accuracy	Precision	Recall
Proposed (Enhanced TL VGG-19+SVM)	98.94	99.59	99.38
Muhammed E.H. et al. [51]	98.30	100.00	96.70
Armando Ugo Cavallo et al. [52]	91.80	--	93.00
Sheetal et al. [53]	94.40	--	94.50
Amira et al. [54]	91.34	91.00	88.33

## Data Availability

The data presented in this study are available online and on request from the corresponding author.

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
