# Peer review of "Multisource Smart Computer-Aided System for Mining COVID-19 Infection Data"

_healthcare, 2022, doi:10.3390/healthcare10010109_

Round 1

Reviewer 1 Report

The manuscript ‘Multisource Smart Computer-Aided System for Mining COVID-19 Infection Data’ by Abou-Kreisha et al. described a Computer Aided Diagnosis (CAD) method for the diagnosis of COVID-19 infection in patients using X-rays and CT- scans of the lungs. The authors not only proposed the development of a hybrid deep leaning algorithm for COVID-19 diagnosis, but also demonstrated its efficacy (high degree of accuracy) in correctly diagnosing COVID-19 when compared to other previously employed methods. The manuscript cannot be accepted in the present form and it requires extensive editing and revision. Therefore, I recommend that the authors resubmit their manuscript after making the following significant changes that are essential for improving their manuscript:

-Please follow strictly Research Manuscript Sections described in instruction to author’s manuscript submission guidelines.

-Proof read the complete manuscript for serious grammatical mistakes and scientific abbreviations including the SARS-CoV-2 extension (it should be Severe Acute Respiratory Syndrome- Coronavirus- 2).

-Combining a large amount of literature can distract readers from the core findings and purpose of the study. Therefore, I would advise the authors to concentrate on, 1) identifying current gaps and how your study addresses them, 2) describing only those methods performed by the authors in the materials and methods section, and 3) presenting only those results obtained by the authors performed, since you are writing a research paper, not a review article.

-In the discussion section, you can compare the results of past studies and illustrate how your results are superior to those of past studies.

Author Response

Dear Reviewer

I am appreciated for your valuable comments that enrich our manuscript and increase the quality of the final version. We have worked on every single comment to enhance and resolve it carefully with regards to your valuable feedback. Next, we have revised, edited, and organized layout of the manuscript using the MDPI editing service. please, find the attached peer review reputable document hoping it fits your expectation. 

Reviewer 2 Report

This paper proposes various neural network models to process X-ray images and determine whether a patient is suffering from COVID-19 or not. The biggest problem is the manuscript is very poorly written and has not been carefully edited at all. There are incomplete sentences everywhere, line breaks in strange locations, and things like “equation1” rather than equation (1). Please edit your manuscripts closer before submitting to a journal.

In this case, the immense number of English errors and strange sentences means I cannot really evaluate the manuscript. It must be submitted to a professional English language service. I include a non-exhaustive list of errors at the end of this review. But really, the entire manuscript needs to be rewritten.

Apart from that, my one major suggestion is that the authors rewrite their methodology section. On the one hand, the description of how neural networks work is much too long. And yet, you leave out essential things. For example, on page 13 you describe details about your implementation in terms of epochs, but you have not defined this elsewhere in the manuscript.

List of English issues: errors or unclear sentences

“diagnosing the covid-19 infection”->diagnosing COVID-19 infections

“in the covid-19 pandemic”->during the COVID-19 pandemic

“The optimal parameter list values were found using Bayesian” - Bayesian what?

“in the twenty-one-century” 

“caused by the new ex-treme acute coronavirus syndrome 2” SARS-CoV-2 is a virus, not a syndrome

“there has been more”->there have been more

“Therefore, the planet is increasingly evolving” this is a strange thing to say

“Image scan is helpful to diagnose COVID-19 for patients that have been exposed to and have severe symptoms of the virus, the outcome of RT-PCR tests [4-6] can still be non-deterministic.” incomplete sentence

“Image scan includes X-rays (CXR), and Computed Tomography images (CT) has proven to be one of the most accurate methods of diagnosis for COVID-19 [7].” strange sentence - should probably be split into two

“Consequently, the COVID-19 diagnosis doesn't use it,” -> it is not usually used in COVID-19 diagnosis

Be consistent: COVID-19 or Covid-19 and X-rays or x-rays

“COV-2 SARS”->SARS-CoV-2

“ARIMA statement model” what is this? I think it should be ARIMA model

“Theoretical papers support the concepts of settled model, prediction models, Bayes-ian network rules, ARIMA statement model, SEIR model, etc.” Unclear sentence, and perhaps models should be plural. Citations necessary.

“Both tools might help doctors diagnose COVID-19 cases more quickly and accurately if combined with them.” delete the last part of the sentence, not necessary.

“So maybe designed to develop computer-based models for predicting, foretelling, analyzing, and distributing COV-2 SARS drugs, allowing ma-chine learning, computer vision, and robotic technology to be applied.” unclear and incomplete sentence

“the COVID-19 visualization of outbreak information” what does this mean?

“In-terestingly, the rapid expansion of deep learning technology confirms usage widely in the health domain” what does this mean? Confirms usage? Do you mean these technologies have been used widely?

“This paper is going to discuss the most recent research in Section 2. Whereas Section 3 presents an overview of the methods and techniques used.” Second sentence is not complete; first is informal.

“COVID-19 measles” what is this?

“Hence the COVID-19 widespread pandemic could constitute a severe threat to global health.” What does this mean? “Hence” usually follows a previous sentence, and is a deduction. And why “could” - of course COVID-19 constitutes a severe threat to health.

“Deep Neural Networks (DNN) or Deep Learning (DL)” ->Deep neural networks (DNN) or deep learning (DL) - they are not proper nouns. Same for “Generative Adversarial Networks (GAN) [40], Convolutional Neural”

“spatial and time dependencies” no line break

“In CNNs, the height and width have shown by m, and r to the channel number or depth is split into the three input components, ? × ? × ? but the input is separated by m and r.” Very unclear sentence.

“equation1” 

Your definition of precision vs recall is quite unclear.

Author Response

(The authors gave the same response as above.)

Round 2

Reviewer 1 Report

I am satisfied with the responses that are provided by the authors. I hereby recommend to publish this manuscript at Healthcare.

Author Response

Dear Reviewer

I am appreciated for your positive feedback

sincerely 

Reviewer 2 Report

The authors have used a professional language service to edit their manuscript. The quality of writing has been improved significantly. 

1) However, the authors have not taken some of my suggestions, and some errors remain that I pointed out the first time.

“by the new severe acute respiratory syndrome coronavirus 2 (SARS-CoV-2).” SARS-CoV-2 is a virus, not a syndrome. I pointed this out the first time.

“The input is separated by m and r instead of the three input compo-nents, ? × ? × ?.” This sentence is still unclear.

The authors have still not defined what an epoch is, for example, only mentioned once in 5.2.2.

2) There are some minor errors left.

“Image scanning include”->Image scanning includes

“Three class label”->three-class label, be consistent

“This section discusses the superiority of the proposed models versus the relative models in recent literature studies” relative->related

3) I have some concerns with the structure of your paper, particularly Sections 1-4. The authors should really restructure the paper. There are several paragraphs that don’t fit.

“Theoretical papers support the concepts of the settled model, prediction models, Bayesian network rules, ARIMA model, SEIR model, etc. A large variety of evidence sup-ports the use of machine learning (ML) algorithms in economic forecasting” this seems not to fit, your paper isn’t about economic forecasting.

“The widespread COVID-19 pandemic constitutes a severe threat to global health. Therefore, most new research has used tools and techniques for tracking COVID-19 and discovering various infection areas to minimize the risk of its spread. Because of the mas-sive quantity of data available every day for COVID-19 infection, spread, detection, deaths, etc., there is a need for big data analytics, storage, and security in NoSQL database management systems [35,36]......” This paragraph could go in the introduction.

“A broad spectrum of computer vision applications has been developed in recent years. The primary goal of such wide use is to imitate the human eye cortex system in form and function by following a hierarchy of feature representation in terms of deep learning and convolutional neural networks (CNNs).....” this paragraph is much too detailed and could be shortened or cut.

Essentially, I recommend that the authors structure it like this:

-introduction to COVID-19 and diagnosis

-background on machine learning and deep learning, including its use in health

-*brief* coverage of previous work, such as AlexNet. 

The detail on neural networks can be drastically reduced.

Author Response

Dear Reviewer

I am appreciated for your valuable comments that enrich our manuscript and increase the quality of the final version. We have worked on every single comment in the second review round to enhance and resolve it carefully with regards to your valuable feedback. please, find the attached peer review reputable document hoping it fits your expectation.
